# Insights into the Structure and Function of TRIP-1, a Newly Identified Member in Calcified Tissues

**DOI:** 10.3390/biom13030412

**Published:** 2023-02-22

**Authors:** Jaison Arivalagan, Amudha Ganapathy, Kalimuthu Kalishwaralal, Yinghua Chen, Anne George

**Affiliations:** 1Brodie Tooth Development Genetics & Regenerative Medicine Research Laboratory, Department of Oral Biology, University of Illinois at Chicago, Chicago, IL 60612, USA; 2Division of Cancer Research, Rajiv Gandhi Centre for Biotechnology, Thiruvananthapuram, Kerala 695014, India

**Keywords:** TRIP-1, exosomes, EIF3i, calcification, MC3T3, preosteoblast, osteogenic, mineralization

## Abstract

Eukaryotic initiation factor subunit I (EIF3i), also called as p36 or TRIP-1, is a component of the translation initiation complex and acts as a modulator of TGF-β signaling. We demonstrated earlier that this intracellular protein is not only exported to the extracellular matrix via exosomes but also binds calcium phosphate and promotes hydroxyapatite nucleation. To assess other functional roles of TRIP-1, we first examined their phylogeny and showed that it is highly conserved in eukaryotes. Comparing human EIF3i sequence with that of 63 other eukaryotic species showed that more than 50% of its sequence is conserved, suggesting the preservation of its important functional role (translation initiation) during evolution. TRIP-1 contains WD40 domains and predicting its function based on this structural motif is difficult as it is present in a vast array of proteins with a wide variety of functions. Therefore, bioinformatics analysis was performed to identify putative regulatory functions for TRIP-1 by examining the structural domains and post-translational modifications and establishing an interactive network using known interacting partners such as type I collagen. Insight into the function of TRIP-1 was also determined by examining structurally similar proteins such as Wdr5 and GPSß, which contain a ß-propeller structure which has been implicated in the calcification process. Further, proteomic analysis of matrix vesicles isolated from TRIP-1-overexpressing preosteoblastic MC3T3-E1 cells demonstrated the expression of several key biomineralization-related proteins, thereby confirming its role in the calcification process. Finally, we demonstrated that the proteomic signature in TRIP1-OE MVs facilitated osteogenic differentiation of stem cells. Overall, we demonstrated by bioinformatics that TRIP-1 has a unique structure and proteomic analysis suggested that the unique osteogenic cargo within the matrix vesicles facilitates matrix mineralization.

## 1. Introduction

Eukaryotic initiation factor (EIF3) plays a vital role in providing a docking site for several EIFs that assemble on the 40S ribosomal subunit, leading to translation initiation [1]. Apart from protein synthesis in eukaryotes, eIF3 was also reported to participate in ribosome biosynthesis [2]. EIF3 in mammals is composed of 13 subunits named EIF3a through EIF3m, among which EIF3a, b, c, g and i are highly conserved in eukaryotes. Human eIF3i is also called p36 or TRIP-1 (Transforming growth factor beta Receptor type II Interacting Protein-1) and contains 325 amino acids. Recently, TRIP-1 has gained interest due to its ability to promote oncogenesis, angiogenesis, and cell proliferation [3,4].

TRIP-1 acts as a modulator of TGF-β activity, which is a potential growth inhibitor for many cell types and induces the expression of several extracellular matrix proteins and adhesion receptors [5]. With respect to the calcification process, TRIP-1 has been suggested to play a regulatory role in osteoblast differentiation by modulating the gene expression of several osteoblastic markers such as Runx2, a master transcription factor for osteoblast differentiation, “early” and “late” markers such as collagen-1, alkaline phosphatase, osteopontin, and osteocalcin that are often associated with the ossification [6,7] process. Higher expression of TRIP-1 mRNA and protein was observed during the early stages of osteogenic differentiation, suggesting its requirement in the osteoblast maturation process [8]. In a recent study, we demonstrated the localization of TRIP-1 in the extracellular matrix (ECM) of bones and at the mineralization front in dentin. In the chondrocytes, TRIP-1 was observed in the proliferating chondrocytes; however, with the development of the epiphyseal growth plate its expression was confined to primary ossification centers. It is quite intriguing how a protein that does not possess a signal peptide could be exported and localized in the ECM. Recently, Ramachandran et al. confirmed the transport of intracellular TRIP-1 to ECM via extracellular vesicles [8].

A functional role for TRIP-1 in the ECM was recently ascertained by nucleation experiments performed on deproteinized and demineralized dentin wafers using varying concentrations of recombinant TRIP-1. Results indicated that TRIP-1 promoted the nucleation and growth of hydroxyapatite crystals [9]. In fact, the binding of TRIP-1 to type-1 collagen, the predominant scaffolding protein in bones and teeth, facilitated hydroxyapatite nucleation.

During the formation of calcified tissues, matrix vesicles (MV) are released by terminally differentiated odontoblasts, osteoblasts, and chondrocytes. These nano vesicular structures are believed to be carriers of calcium phosphate mineral that is transferred to the organic protein template such as collagen during skeletal tissue formation. Chaudhary et al. reported that the higher concentration of phosphate in the culture media induced pre-osteoblast MC3T3-E1 cells to secrete MVs [10]. Intriguingly, we observed TRIP-1 overexpressing MC3T3-E1 cells secreted extracellular vesicles several folds higher than that of the control cells when observed under FESEM^7^. The number of vesicles was directly proportional to the amount of ECM proteins exported to the matrix.

It is still unclear how an intracellular translation initiation factor can function in angiogenesis as well as in mineralization. Therefore, we examined the multifunctional role of TRIP-1 at a molecular level using bioinformatics tools. This is a strategy currently employed to elucidate the function of a protein in-silico using several freely available web-based bioinformatics tools. In this study, we integrated several bioinformatics approaches to elucidate the structure and identify functional domains and interacting proteins to understand its various functions in the extracellular matrix. Further, we examined if TRIP-1 influenced the osteogenic cargo of matrix vesicles to promote matrix mineralization by carrying out a label-free proteomic quantification approach on matrix vesicles isolated from both control and TRIP-1-overexpressing pre-osteoblast MC3T3 cells.

## 2. Results

### 2.1. Phylogenetic Analysis:

The phylogenetic analysis of 61 EIF3i sequences (Figure 1) from organisms belonging to different families verified the conservation of EIF3i as reported earlier [11]. The sequences of EIF3i are very similar in length, showing the evolutionary relationships among biological species. N-terminal and C-terminal regions are highly conserved in plants and animals.

### 2.2. Conserved Domains and PTMs

EIF3i human protein contains 325 amino acids. The SMART tool retrieved five WD40 domains and four different types of post-translational modifications in the sequence of TRIP-1. The highest number of PTMs was in ubiquitination sites, which were dispersed throughout the sequence. Two nitrosylation and two acetylation and putative phosphorylation sites were also reported (Figure 2).

### 2.3. Tertiary Structure and Ion Binding Sites

Modeling TRIP-1 using I-TASSER retrieved the top five models; the closest model selected was based on the C-score and TM-score. The C-score is assigned by the I-TASSER tool which often ranges from −5 to 2 and the higher value signifies the structural model with high confidence. Similarly, a higher TM-score signifies a closer structural similarity and such proteins exhibit similar functions [12]. The results presented showed that TRIP-1 forms a seven-bladed ß-propeller structure, and it was modeled using rabbit Eif3i protein (RCSB PDB ID, 5K0Y: chain T) (Figure 3). COFACTOR and COACH (tools that predict function based on I-TASSER generated model) suggested that Wdr5 and G-protein ß subunit (GPSß) proteins have a similar structure as TRIP-1 and therefore could have similar functions.

### 2.4. Data Mining:

#### 2.4.1. BioGRID Database

The biological network paradigm was used to investigate the interacting partners of TRIP-1 using BioGRID (Biological General Repository for Interaction datasets). As a result, 173 protein–protein interactions with 120 interactors were identified (Appendix A). Protein interactions of TRIP-1were annotated in BioGRID. GO biological processes identified were cellular protein metabolic process, gene expression, translation, and translational initiation; the GO molecular functions identified were protein binding and translation initiation factor activity; the GO cellular components identified were, cytosol, eukaryotic translation initiation factor 3 Complex, and extracellular vesicular exosome.

#### 2.4.2. Ingenuity Pathway Analysis

Molecular network analysis modeling interactions between TRIP-1, collagen-1-alpha-1 (Col1A1) and DMP-1 performed by Ingenuity Pathway Analysis (http://www.ingenuity.com/products/ipa, accessed on 1 April 2019) resulted in the 15 shortest path (one intermediate node) interactions (Figure 4a). Increasing another intermediate node expanded the number of interactions to several folds, i.e., totally, 4726 interactions. Of these, 52 interactions contain TGFR2 (TGF beta receptor 2) as the first node (Appendix A). Enrichment of TRIP-1 in the network plot did not change any of the interactors; however, enrichment of Col1A1 and DMP-1 resulted in modulation of 27 and 29 molecules, respectively, in the pathway (Appendix A). The shortest path connecting TRIP-1, calcium phosphate, durapatite (hydroxyapatite) and basic calcium carbonate contains at least two nodes and 54 interactions (Figure 4b).

### 2.5. Nanoparticle Tracking Analysis (NTA) and Transmission Electron Microscopy (TEM)

Matrix vesicles isolated from control and MC3T3-TRIP-1OE cells were subjected to particle size analysis by NTA. The results from NTA analysis showed that the concentration of MV’s isolated from TRIP1-OE cells were higher than that of MC3T3. Moreover, MVs of MC3T3-TRIP-1OE ranged in size from 110 to 188 nm while the size of MVs from MC3T3 had sizes from 99 to 181 nm (Figure 5a). Representative TEM images (Figure 5b) of MVs demonstrate vesicles containing a lipid bilayer with sizes ranging from ~100–150 nm.

### 2.6. Proteomic Analysis:

Out of 330 proteins identified from MVs isolated from MC3T3 and MC3T3-TRIP-1OE cells, 142 proteins were common to both cell types (Appendix A). However, applying the minimum two-fold change cut-off filter identified only five proteins, such as major vault protein, Ras GTPase-activating-like protein (rasGAP), catenin alpha-1, interferon-induced transmembrane protein 2 and 3 (IFITM-2 and -3), highly expressed in TRIP-1OE MVs. In total, 153 proteins were found specific to MC3T3 and 15 specific to MC3T3-TRIP-1OE. Out of the 153 proteins, nearly a third of the proteins were related to the metabolic process, 16% were cellular organizers or molecules involved in biogenesis, 13% were biological regulators and 12% localizers (Figure 6). Interestingly, some proteins that are specific to MC3T3-TRIP-1OE are very important in mineralization. These include, to name a few, collagen 1 alpha, fam3, Vwa5a, fibrillin-1, pantetheinase, etc. (Appendix A).

### 2.7. Functional Assay to Demonstrate the Effect of TRIP1OE-MVs on Stem Cell Differentiation

Matrix vesicles isolated from MC3T3-E1 and MC3T3-TRIP1OE cells were evaluated for their potency to induce mineralization-related gene response in dental pulp stem cells (DPSCs). Gene expression analysis of six selected gene transcripts by quantitative PCR showed that MC-MVs upregulated the expression of RUNX2, OCN, BMP2, BMP6, HIF-1alpha, and VEGF under normal growth and differentiation conditions at 4 and 24 h (Figure 7). However, exposure to TRIP1OE-MVs had a profound influence on the expression levels of Runx2, BMP6, OCN and BMP2 only under differentiation conditions while the expression levels of HIF1 alpha and VEGF stayed at similar levels to MC-MVs. (Figure 7e,f). These results confirm that TRIP-1 can modulate the contents of the matrix vesicles to facilitate osteogenic differentiation of stem cells.

## 3. Discussion

Analyzing the sequences of EIF3 subunit-i showed that they are conserved in all classes of eukaryotes including Mammalia. Our results agree with the previous phylogenetic analysis carried out on EIF3i sequences from 12 organisms, which includes Trypanosomatids and lower eukaryotes [11]. Majority of EIF3i sequences show short branches indicating the more rapid accumulation of substitutions without changing the phylogenetic relationships. In animals, *C. briggsae*, *C. elegans,* and *B. malayi* displayed long branches. In general, the sequence determines structure and structure determines molecular function [13]. Hence it can be stated that the homologous proteins share similar functions [14], suggesting the functional conservation of EIF3i in eukaryotes.

Predicting the function of the protein solely based on the sequence similarity can be unreliable in some cases, since a common evolutionary origin does not guarantee functional conservation of paralogs and the more distant the evolutionary relationship, the less reliable the transfer [15]. Usually, in such cases, looking into short, conserved motifs using PSI-BLAST might refine and improve the homology-based functional predictions. However, in the case of EIF3i, homologous protein search in addition to PSI-BLAST did not predict any functions other than translation initiation. This encouraged us to search for the functional regions, i.e., conserved domains present in the EIF3i human sequence.

TRIP-1 does not contain any structural motifs other than the identified WD40 domains. WD40 is a domain made of ≈40 amino acid residues, which form a beta-propeller structure and are common in eukaryotes than in prokaryotes. They are involved in signal transduction pathways that are unique to eukaryotes. It is not clear if WD40 domain repeats are of ancient origin or if it is a eukaryotic invention borrowed by some prokaryotes via horizontal gene transfer [16]. The cell surface protein in prokaryote contains a YVTN-repeat seven-bladed beta propeller structure, which is similar to eukaryotic cell surface receptors which also contain YVTN-repeats with seven-bladed beta propeller structures, suggesting a common ancestral relationship between the prokaryotes and eukaryotes [17]. The SMARTnr database alone harbored 799 (0.8% of total human proteins in the database) human proteins containing 3943 WD40 domains (0.72% of total human domains in the database). Of the 799 proteins, majority of the proteins’ functions could be related to cell signaling and others are involved in a variety of cellular functions except exhibiting any kind of catalytic function [18]. It is known that the WD structural motif provides protein–protein binding surfaces for reversible protein complex formation [16], e.g., several trimeric G-proteins and their downstream complexes mainly involving MAP Kinase cascades. In humans, many diseases are associated with the impaired WD repeat-containing proteins. These include, to name a few, G-protein beta-3 subunit splice variant in hypertension, mutation in AAAS protein causing Triple-A syndrome, and the role of WD11 and STRAP proteins in glioblastoma and carcinogenesis [19,20,21]. Thus, it is evident that WD40 is an important structural motif that is highly conserved during evolution. Published reports show the different functions attributed to proteins containing the WD domain. Therefore, the question arises of whether the multifunctional WD domains possess functions related to biomineralization.

Most proteins perform biological functions via interactions with other proteins. Predicting the function of a protein from their interacting partner is reasonable since 70–80% of the proteins share the function of at least one interacting partner [22]. Therefore, we sought to identify the function of TRIP-1 by identifying its interacting partners;120 interacting partners of TRIP-1 that were identified include several proteins associated with extracellular vesicles, e.g., cullin-3, intersectin-2, and exosome component-6. In fact, TRIP-1’s association with extracellular vesicles was demonstrated in our previous work [7]. Few lectins were found to interact with TRIP-1. Lectins are widely found in many biomineralized structures including sea urchins (SM proteins) [23], molluscs (perlucin of abalone nacre) [24], and eggshells (struthiocalcin of ostrich eggshell) [25]. In sea urchin spicules, lectins were transported in vesicles from the cell surface and incorporated into the mineralizing matrix of developing spicules [26]. In humans, osteoblast-derived C-type lectin was found to inhibit the formation of osteoclasts [27]. We propose that TRIP-1 might interact with lectins and facilitate biomineralization.

It is widely accepted that Col1A1 and DMP-1 are important matrix molecules in biomineralization. Ingenuity Pathway Analysis was used to identify the molecular interaction network of TRIP-1 with DMP-1 and Col1A1. The shortest path with one intermediate node between Col1A1, DMP-1, and TRIP-1 identified mainly nuclear and membrane proteins including TGFBR2 (TGF beta receptor-2) and two extracellular proteins, angiotensinogen (AGT) and vitronectin (VTN) (Figure 4a). Since no studies have demonstrated the association of TRIP-1 with biomineralization, the enrichment of TRIP-1 in silico in the interactome predicted by IPA did not influence the composition of upstream and downstream molecules in the interactome. Contrarily, enrichment of DMP-1 and Col1A1 yielded several up- and downstream molecules (Appendix A) including TRIP-1, demonstrating that the functional role of TRIP-1 might be like that of DMP-1 and Col1A1, i.e., regulation of matrix mineralization processes. Earlier work from our group demonstrated the association of TRIP-1 with Col1A and its function in regulating hydroxyapatite nucleation and growth [9].

Wdr5 and GPSß are structurally similar to TRIP-1 and it could be extrapolated that proteins with similar structures could have similar functions [12]. Wdr5, a protein belonging to WD family of proteins, contains seven WD40 domains homologous to TRIP-1 and plays a key role in the differentiation of osteoblasts and chondrocytes [28]. Wdr5 was also identified in the matrix of alveolar bone [29]. Functional annotation of Wdr5 resulted in histidine methyl transferase activity; however, the GO: 0001501 pointed out its involvement in the development of bone and cartilage. GPSß is very similar to TRIP-1 and contains seven WD40 domains and forms a ß-propeller structure. It interacts with calmodulin and increases the alkaline phosphatase activity, a requirement in mineralization. In pearl oyster and Zhikong scallop, it is confirmed that GPSß is essential for shell formation [30,31]. However, interactions of these proteins with the mineral have not been confirmed. Intriguingly, in our published study, we demonstrated the binding of recombinant TRIP-1 with calcium phosphate leading to hydroxyapatite nucleation and growth in vitro [8].

Two SGE regions in TRIP-1 resemble the SXE motif, which has been shown as a common sequence in the secretory calcium-binding phosphoprotein (SCPP) family and many small integrin-binding ligand N-linked glycoproteins (SIBLING) involved in the mineralization of bones and teeth [32]. It has also been identified as a functional motif in the protein odontogenesis-associated phosphoprotein (ODAPH), which functions in enamel mineralization and is mutated in various forms of amelogenesis imperfecta [33]. Although IPA could detect several indirect interactions of TRIP-1 with calcium phosphate (Figure 4b), functional studies are required to confirm the mineral-binding properties of these in-silico identified domains.

Phosphorylation of serines and threonines are major post-translational modifications in several of the non-collagenous proteins that are responsible for the assembly of a calcified matrix in bone and dentin [34,35]. Enzymes such as casein kinase (CK) have been reported to phosphorylate matrix proteins such as phosphophoryn, bone sialoprotein, dentin matrix protein 1, and osteopontin [36]. Examining the sequence of TRIP-1 shows 12 potential phosphorylation sites, of which five sites are predicted to be substrates for CK1 and CK2 enzymes (CK1- 219T, 301S; CSK2- 167S, 218 T, 302S) (Appendix A) indicating the potential for TRIP-1 to be phosphorylated as other acidic non-collagenous proteins in mineralized matrices.

The role of matrix vesicles in biological calcification has been well studied. These nanosized membrane invested extracellular particles are believed to initiate mineralization in dentin, growth plate cartilage, and developing bone. They are implicated not only in events related to bones and teeth but also in the pathogenesis of certain diseases in which too little or too much calcification occurs. Cells presumably exert ultimate control over the initial mineral induction events by dictating what components are packaged within the vesicles. Therefore, we examined if TRIP-1 influenced the contents of the matrix vesicles isolated from preosteoblast MC3T3-E1 and MC3T3-TRIP-1OE cells. Proteomic analysis showed five proteins were highly expressed in MVs of MC3T3-TRIP1-OE (log2fold change > 2) (see Appendix A). Expression of these proteins is quite intriguing since most of the previous literature discusses their role either as signaling molecules or immune-related functions [37,38,39,40].

Fifteen proteins were found to be specific to MC3T3-TRIP1-OE MVs (Appendix A). Vwa (Von Willebrand factor A) domain-containing proteins were identified and they are frequently reported as an adhesion molecule [41] and might glue mineral crystals [42] during calcified tissue formation. Other identified proteins include those such as Fam3C, a cytokine-like growth factor, in which a single nucleotide polymorphism in *Fam3c* gene affects the bone mineral density in humans [43]. Results with Fam3C knock out mouse confirmed its role in osteoblast differentiation and bone homeostasis [44]. Fibrillin-1 and -2 are structural components of extracellular micro fibrils. They regulate bone formation by modulating endogenous TGF-β and BMP signaling [45]. Vanin-1 pantetheinase regulates chondrogenesis via glutathione metabolism and is critical for accelerated chondrogenesis [46]. S100A11 was specifically identified in TRIP-1 OE cells. An earlier study using MVs from human osteosarcoma Saos-2 cells showed the presence of S100A4 and S100A6 but not S100A11 [47]. Moreover, they suggested that S100A11 may not be associated with the initiation of mineralized matrix formation by MVs. It is well established that S100 family of proteins (including 24 proteins) are small EF-hand Ca^2+^-binding proteins that exhibit bothintracellular and extracellular functions [48]. In MC3T3 MVs, annexins 1, 2, 4, 5, 6, 7, and 11 were identified. In fact, they are present in equal amounts in both normal and TRIP-1OE cell types except annexin 7, which is present only in TRIP1-OE MVs. Annexins are Ca^2+^ and lipid-binding proteins, which are involved in Ca^2+^ homeostasis in osteoblasts and in extracellular MVs [49]. All these proteins found unique to TRIP1-OE cells indicate that TRIP-1 drives mineralization by influencing the secretion and export of ECM proteins via MVs. These proteins are interesting starting points for further in-depth studies to define the function of TRIP-1 in matrix vesicles.

## 4. Conclusions

In summary, characterization of the structure of TRIP-1 provides valuable insight into its role in the ECM of calcified matrices. TRIP-1 does not possess the characteristic amino acid signature and structure of classical calcium-binding acidic proteins involved in mineralization. Prediction of its function using several bioinformatics tools suggest that TRIP-1 is a multifaceted protein and confirms our initial observation regarding its mineral nucleating role in the ECM. Identifying the composition of matrix vesicles produced by mineralization-competent cells overexpressing TRIP-1 suggests that the cargo packaged in TRIP1OE-MVs is influenced by TRIP-1. Further, these molecular components stimulated an osteogenic response in dental pulp stem cells. Identification of new proteins by proteomics in TRIP1-MV suggests that calcification is a carefully orchestrated process involving several proteins that are yet to be identified. Overall, this study facilitates our understanding on the role of matrix vesicles and the importance of matrix-mediated signaling in biological mineralization. Such knowledge could translate into the development of therapies for the treatment of calcification disorders.

## 5. Materials and Methods

### 5.1. Phylogenetic Analysis:

Several EIF3I protein sequences belonging to different taxonomic classes were obtained from http://uniprot.org; accessed on 1 March 2019. Manually annotated and reviewed samples were selected for the analysis. In total, 61 protein sequences were aligned using a multiple sequence alignment tool MUSCLE 3.8.31 followed by curation by Gblocks 0.91b, which eliminates poorly aligned positions and divergent regions. The phylogenetic tree was constructed using the maximum-likelihood method in PhyML 3.1. Shimodaira–Hasegawa approximate likelihood ratio test (SH-aLRT) was carried out to evaluate the reliability of internal nodes. The tree was visualized and annotated using publicly available ETE Toolkit tree viewer (http://etetoolkit.org) accessed on 1 March 2019.

### 5.2. Conserved Domain Prediction

The amino acid sequence of EIF3I was analyzed in Simple Modular Architecture Research Tool (SMART; http://smart.embl-heidelberg.de) accessed on 2 March 2019 to retrieve the conserved domains. PTM code version 2, a web resource predicted the post-translational modifications and their functional associations.

### 5.3. Interacting Partners and Post-Translational Modifications

The TRIP-1-interacting partners were generated by searching against BioGRID interaction repository 3.5. DiANNA 1.1 was used to predict the disulphide bonds formed by cysteines in the sequence. It determines the cysteine species (free cysteine, half-cystine or ligand-bound) by using a support vector machine (SVM) with degree 2 polynomial kernel for the spectrum representation and the disulphide connectivity is predicted using a state-of-the-art method involving a novel architecture neural network. Phosphorylated residue and kinase specificity were predicted using PhosphoSite Plus^®^ and NetPhos 3.1 respectively.

### 5.4. Ingenuity Pathway Analysis

The molecular interaction networks were generated using Ingenuity Pathways Analysis (IPA^®^, QIAGEN, Inc. Redwood City, CA, USA). Molecule activity predictor (MAP) in IPA was used to enrich a molecule of interest in a pathway, in silico, to identify the consequent up and downstream effects.

### 5.5. Matrix Vesicle Isolation

Mouse calvaria osteoblast precursor cell lines (MC3T3-E1) and TRIP-1-overexpressing MC3T3 (MC3T3-TRIP-1OE) cell lines were grown in alpha Minimum Essential Medium (α-MEM) containing 10% fetal serum and 1% Antibiotic-Antimycotic until 90% confluence was reached. For MV isolation, the 90% confluent cell cultures were washed several times with 1X PBS to remove serum-derived vesicles and the cells were grown in serum-free α-MEM medium for 14 days. The medium was replaced with fresh medium every two days. The MVs were released from the extracellular matrix by adding 2 mL of collagenase (1.5 mg/mL) to each plate. MVs were separated from cells and other cellular debris by centrifuging at 2000 rpm for 10 min and 10,000 rpm for 30 min. The supernatant obtained was centrifuged at 120,000 rpm for 2 h to pellet MVs.

### 5.6. Nanoparticle Tracking Analysis (NTA)

Part of the MV sample was solubilized in 50 uL PBS for NTA and TEM analysis. The abovementioned sample was diluted with sterile PBS in a 1:200 ratio and analyzed using NanoSight NS 300 instrument (Malvern Instruments Limited; Malvern, Worcestershire, UK). Three 30 s frames were captured for each sample to ensure of the size and number of MVs. Nanosight NTA software Version 3.2 Dev Build 3.2.16 was used to determine the average distribution and concentration of MVs.

### 5.7. Transmission Electron Microscopy (TEM)

MV sample was applied on a copper grid (mesh-400 lines/inch) and air dried; 10% formalin was used as a fixation agent. After fixation the grid was washed several times and dried completely. The grids were imaged using TEM (JEM-1220, JEOL) at 300 Kv.

### 5.8. Real-Time PCR

Dental pulp stem cells (DPSC) were grown to confluence before treating them with matrix vesicles isolated from MC3T3 and MC3T3-TRIPOE cells cultured in growth and differentiation media (growth media supplemented with 10 mM β-glycerophosphate, 100 mg/mL ascorbic acid and 10 nM dexamethasone). Total RNA from MV treated DPSC cells was isolated using TRIzol™ (Invitrogen™) reagent and the cDNA was synthesized using Maxima First-strand cDNA synthesis kit (Thermo Scientific™) according to the manufacturer’s instructions. Specific primers for the *BMP-2*, *BMP-6*, *RUNX-2*, *OCN*, *HIF1-A*, *VEGFA,* and *GAPDH* are listed in Appendix A. Quantitative PCR was carried out in StepOnePlus Real-Time PCR system (Applied Biosystems, Waltham, MA, USA). Target gene expression values were normalized with housekeeping gene GAPDH. Fold change was calculated using 2^−∆∆Ct^ method. The expression value of DPSC cells treated (t = 0) with exosomes from MC3T3 cells cultured in growth media was set as 1 for ∆∆Ct calculation.

### 5.9. Proteomic Analysis

#### 5.9.1. Peptide Fractionation

The MVs were solubilized using 5% SDS in 1X RIPA buffer. From each sample, 25 μg of proteins was taken and processed following standard S-Trap Micro protocol [50]. Proteins underwent tryptic digestion at a 1:10 ratio (trypsin:protein) at 47 °C for 1 h, and resulting peptides were eluted from the spin column, dried down in the speed vac, and reconstituted in 50 μL of 20 mM ammonium hydroxide and subjected to high-pH HPLC fractionation using Waters Xbridge C18 column (3.5 μm, 4.6 × 250 mm). Mobile phase A was 20 mM ammonium hydroxide in water at pH 10; mobile phase B was 20 mM ammonium hydroxide in acetonitrile at pH 10. Peptides were eluted out using a 60 min gradient program where mobile phase B increases from 1% to 60%. Fractions were collected every min throughout the gradient. As a result, 60 fractions were collected. These samples were then combined to a total of 10 samples, dried, and reconstituted into 15 μL of 5% acetonitrile in water with 0.1% formic acid individually. One μL of each sample was injected for LC-MS/MS analysis.

#### 5.9.2. Mass Spectrometry

Samples were analyzed by a Thermo Orbitrap Velos Pro mass spectrometer coupled with an Agilent nanoflow LC system. Peptides were eluted on Agilent Zorbax 300SB-C18 column (3.5 um, 0.075 × 150 mm) via 60 min gradient LC program where mobile phase B increases from 7% to 40%. Mobile phase A: water with 0.1% FA; mobile phase B: ACN with 0.1%FA. Data were acquired by a DDA method (data-dependent acquisition); higher-energy collisional dissociation fragmentation was performed for the top-12 peaks in each MS scan.

### 5.10. Protein Identification and Quantitation

MS/MS spectra obtained from the mass spectrometer were searched in MASCOT (Ver 2.5.1, Matrix Science., London, UK) against a UniProt mouse database. The following parameters were used in the database search: carbamidomethylation of cysteine as a fixed modification and oxidation of methionine and de-amidation of aspartic acid and asparagine as variable modifications. The mass tolerance for MS and MS/MS experiments was set as 20 ppm and 0.3 Da, respectively. Further validation was carried out in Scaffold (Ver 4.8.4, Proteome Software Inc., Portland, OR, USA). Peptide and protein identifications were accepted only if the FDR was less than 1% and contained at least 2 peptides. Proteins that contained similar peptides and could not be differentiated based on MS/MS analysis alone were grouped to satisfy the principles of parsimony. The identified proteins were relatively quantified in Scaffold based on their normalized total spectra. Proteins were classed using PANTHER classification system.

### 5.11. Statistical Analysis

Data were presented as mean ± standard deviation and significance was declared at *p* < 0.05. One way ANOVA was used to determine the difference between the groups. Post-hoc Tukey’s honestly significant difference test was used to find whether the means were significantly different between the groups.

## Figures and Tables

**Figure 1 biomolecules-13-00412-f001:**
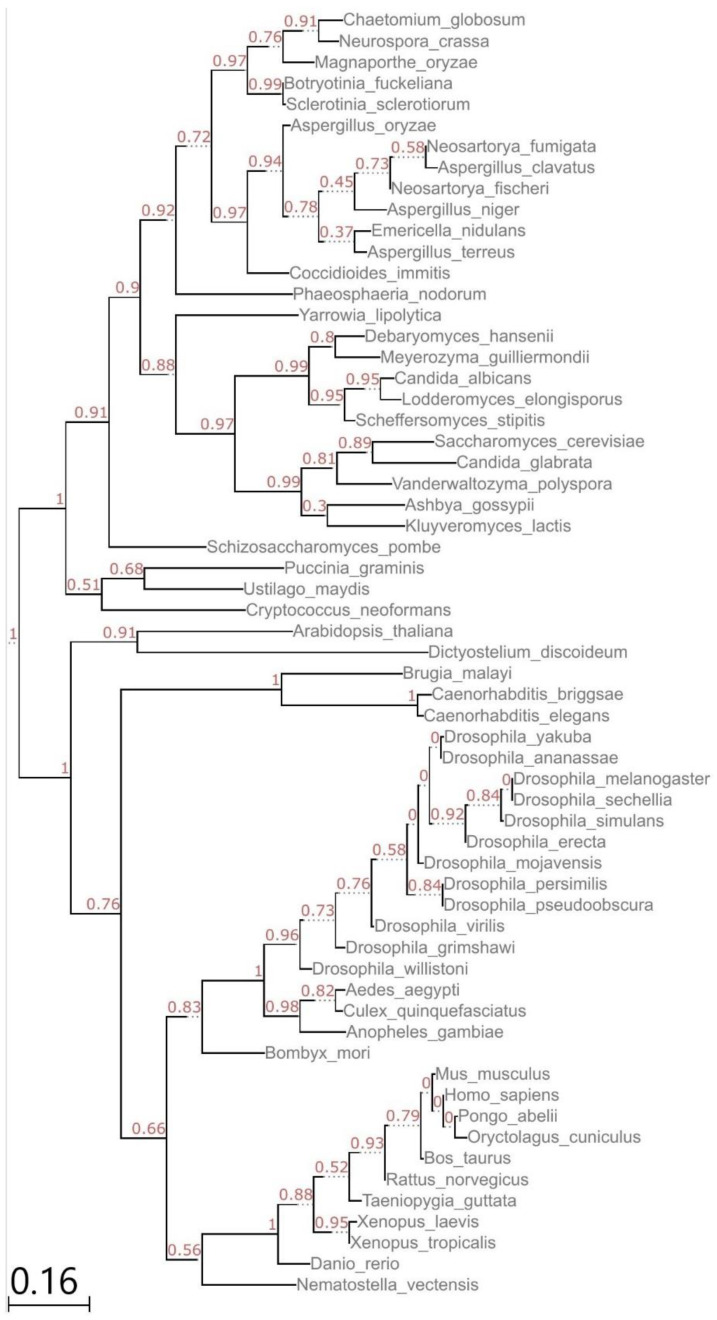
Phylogenetic Tree of EIF3i sequences. Tree constructed from sequence alignment using full-length EIF3i (TRIP-1) protein sequences from 61 eukaryotes.

**Figure 2 biomolecules-13-00412-f002:**
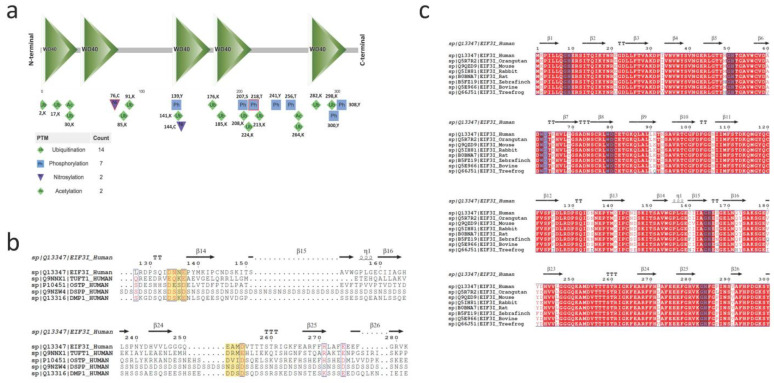
Functional domains and PTM sites present in TRIP-1. (**a**) The conserved domain search in the sequence of TRIP-1 using the SMART tool. Possible post-translational modifications (PTMs) are denoted in different colors and symbols. The number of PTMs is represented beneath the diagrammatic representation of the protein. The complete list of phosphorylation sites from PhosphoSite Plus^®^ is provided in Appendix A. (**b**) Sequence alignment of human EIF3i with other mineralization proteins. (**c**) Sequence alignment of conserved regions of EIF3i of different animals.

**Figure 3 biomolecules-13-00412-f003:**
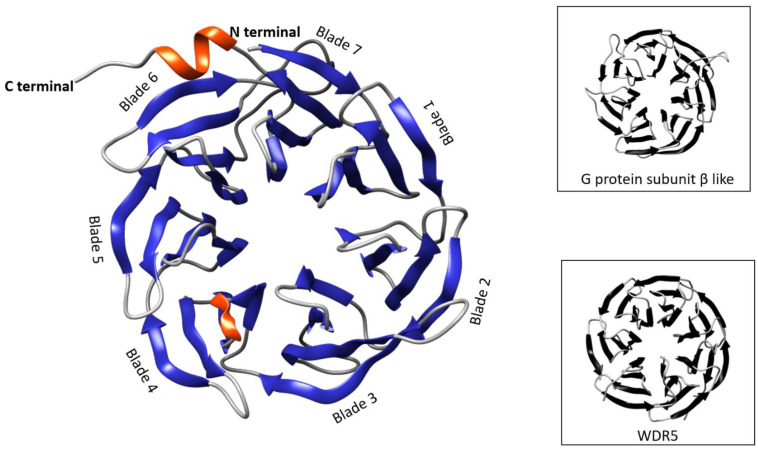
Secondary structure of TRIP-1. The structure of TRIP-1 predicted by I-TASSER revealed several beta sheets conforming to the seven-bladed beta-propeller structure (grey-loop, blue-beta sheets, and orange-alpha helix). The figures in the box are the structures of GPSß (4JSP:D) and Wdr5 (RCSB PDB ID, 3UVL: A).

**Figure 4 biomolecules-13-00412-f004:**
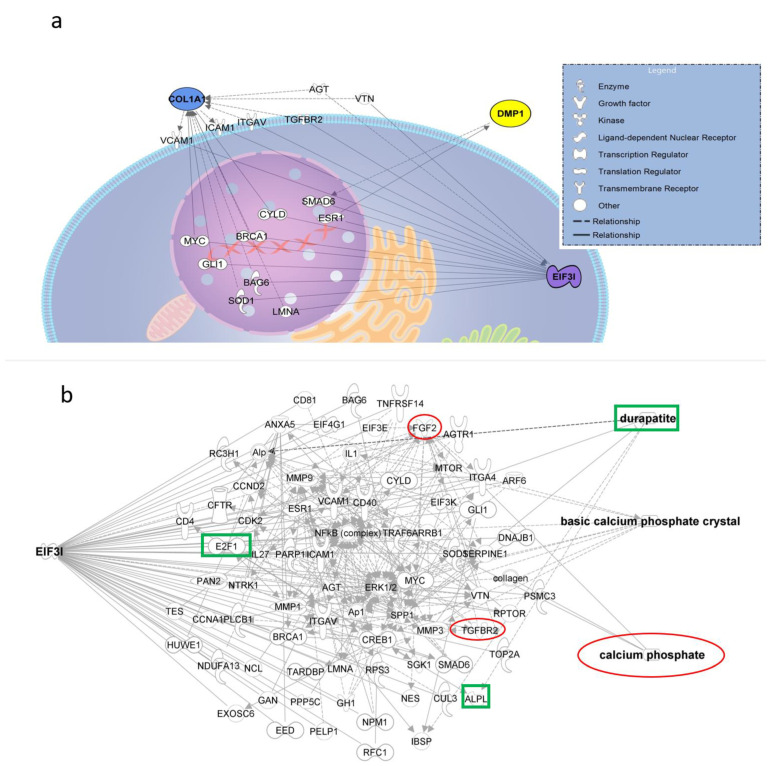
Network plot generated using IPA. (**a**) Molecular network analysis showing the interactions between DMP-1, TRIP-1, and Col1A1 using Ingenuity Pathway Analysis (shortest path with one intermediate node), for example, ‘Col1A1–BRCA1–TRIP-1’ and ‘DMP-1–SMAD6–TRIP-1’. (**b**) Interaction of TRIP-1 with Durapatite (hydroxyapatite), calcium phosphate, and basic calcium phosphate crystal (shortest path with two intermediate nodes). Examples: ‘TRIP-1–TGFBR2–FGF2–calcium phosphate’ and ‘TRIP-1–E2F1–ALPL–durapatite’s.

**Figure 5 biomolecules-13-00412-f005:**
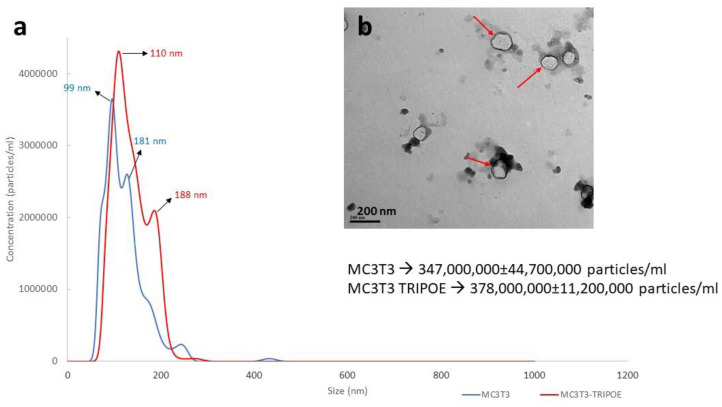
Characterization of the size and concentration of the purified matrix vesicles. (**a**) Nanoparticle Tracking Analysis showed the particle size distribution and concentration of MVs isolated from MC3T3 (blue) and TRIP-1OE (red) cells. X-axis denotes the size (nm) and Y-axis denotes the concentration of MVs (particles/mL). Particle size of MC3T3 MVs ranged from 99 to 181 nm and MC3T3-TRIP-1OE MVs ranged from 110 to 188 nm (**b**) Representative TEM image of MVs of TRIP-1OE cells. The size of the matrix vesicles ranged from 100–200 nm.

**Figure 6 biomolecules-13-00412-f006:**
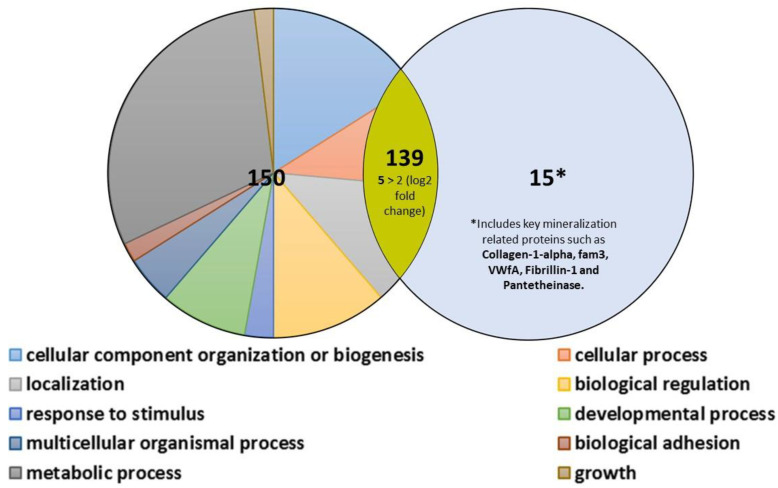
Proteomic signature in MC3T3-E1 and MC3T3-E1-TRIP-1OE matrix vesicles as identified by proteomics analysis. The Venn diagram represents the comparison of the matrix vesicles proteins isolated from MC3T3 (**left**) and MC3T3-TRIP-1OE cells (**right**). In total, 139 proteins were common to both matrix vesicles of which 5 were overexpressed in the vesicles from TRIP-1OE; 15 proteins involved in mineralization were unique to TRIP-1OE cells. Proteins specific to MC3T3 are classed based on their biological process using the PANTHER classification system and depicted in the figure label.

**Figure 7 biomolecules-13-00412-f007:**
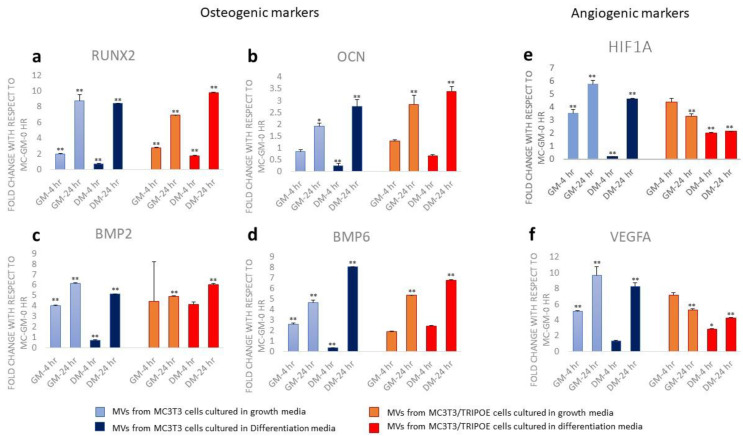
Gene expression analysis of DPSCs stimulated by MC-MV and TRIP1 OE-MV. Quantitative PCR analysis of gene expression of RUNX2, OCN, BMP2, BMP6, HIF-1alpha, and VEGF under normal growth and differentiation conditions at 4 and 24 h after treatment with MVs. *p*-value < 0.05 was considered significant. (GM: growth media, DM: Differentiation media). * *p* < 0.05, ** *p* < 0.01.

## Data Availability

The proteomics and PCR datasets generated during and/or analyzed during the current study are available by request from the corresponding author.

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
