# Peer review of "Insights into the Structure and Function of TRIP-1, a Newly Identified Member in Calcified Tissues"

_biomolecules, 2023, doi:10.3390/biom13030412_

Round 1

Reviewer 1 Report

The manuscript entitled "Insights into the structure and function of TRIP-1, a newly identified member in calcified tissues" is focused on eIF3i (TRIP1) influence on the proteomic composition of matrix vesicles derived from osteoblast precursor cells. The eIF3i is a constitutive part of the translation initiation factor eIF3, however, it has additional non-canonical (non-translation) functions within the cell. The molecular functions of eIF3i are poorly understood. The understanding of its functions is of great importance due to eIF3i involvement in cell differentiation and tumorigenesis. Despite the actuality of the subject, the presented manuscript has serious flaws. 

(1) The first part of the manuscript including phylogenetic, conserved domain analysis, PTMs, and tertiary structure predictions is not novel. Nearly the same results have already been published (for review see doi.org/10.1016/j.canlet.2020.12.009). Moreover, the tertiary structure of TRIP1 is solved for isolated protein and in complex with parts of other eIF3 subunits. 

(2) The cell experiments were performed in control and TRIP1-overexpressed cells. The manuscript doesn't contain any controls regarding the expression of the studied protein, like Western blot.  

(3) The main mass-spectrometry experiment, possibly, made in one replicate. There is no information about replicates and the reproducibility of the analysis. However, the protein composition of matrix vesicles should depend on cell proliferation, their apoptosis status, and confluency and may vary between replicates. That's why it's crucial to repeat the experiment several times and analyze only reproducible proteins. Also, it's important to check the cell viability and proliferation upon TRIP1-overexpression.

(4) Another important issue regarding mass-spectrometry experiments is that there is no experiment on intracellular protein composition. The protein composition of matrix vesicles (MV) strongly depends on intracellular protein level and should be analyzed simultaneously with such data.

The manuscript also contains a lot of minor issues that should be addressed:

(1) No replicates and statistical analysis in Nanoparticle tracking analysis and TEM imaging.

(2)In the qPCR experiment statistical analysis did not perform for GM vs GM-OE pairs, which are discussed in the text.

(3) Figure 7. The presentation of the data is not very demonstrative, especially in the case of decreasing RNA amount. The y-axis should be logarithmic for clarity.

(4) Figure 3 is in the wrong place in the manuscript.

(5) Figure 1. There isn't a sequence alignment that is discussed in the text and figure capture.

(6) Phosphosite Plus prediction of PTMs. These resource does not contain predictions, it accumulates information about PTMs which were detected in high-throughput or low-throughput experiments.

(7) Figures 2, 4, 6. The figure text is so small. It's impossible to read it.

Author Response

We immensely thank the reviewers and the editor for reviewing our manuscript titled "Insights into the structure and function of TRIP-1, a newly identified member in calcified tissues" and for their valuable and constructive comments. The reviewer’s constructive criticisms have helped us in improving the overall quality of the manuscript. As per the editor’s and reviewers’ constructive comments, the corrections were made in the manuscript keeping the track mode on, and submitted for your kind perusal. We hopefully believe that we have addressed all the comments mentioned by the reviewers carefully and precisely. We wish to thank the reviewer for his invaluable comment that has strengthened our manuscript.

The manuscript entitled "Insights into the structure and function of TRIP-1, a newly identified member in calcified tissues" is focused on eIF3i (TRIP1) influence on the proteomics composition of matrix vesicles derived from osteoblast precursor cells. The eIF3i is a constitutive part of the translation initiation factor eIF3, however, it has additional non-canonical (non-translation) functions within the cell. The molecular functions of eIF3i are poorly understood. The understanding of its functions is of great importance due to eIF3i involvement in cell differentiation and tumorigenesis. Despite the actuality of the subject, the presented manuscript has serious flaws. 

  • The first part of the manuscript including phylogenetic, conserved domain analysis, PTMs, and tertiary structure predictions is not novel. Nearly the same results have already been published (for review, see doi.org/10.1016/j.canlet.2020.12.009). Moreover, the tertiary structure of TRIP1 is solved for isolated protein and in complex with parts of other eIF3 subunits. 

We appreciate the reviewer for pointing out similar results published in the previous work. We didn’t use any of their results and all the analyses were performed by us using several bioinformatics tools. Furthermore, we carried out phylogenetic analysis using the different organism sequences collected from the protein repositories. Regarding conserved domain analysis and PTMs, we have just reported the previous findings that were listed in the PTM database, and we haven’t predicted any. We modified these sentences in the text. For the tertiary structure prediction, we used one of the precise tools to compare the tertiary structure of similar proteins to predict the other possible functions of TRP1.

  • The cell experiments were performed in control and TRIP1-overexpressed cells. The manuscript doesn't contain any controls regarding the expression of the studied protein, like a Western blot.  

We agree with the reviewer’s comments. We have used the same cells in our earlier study, and it has all the data regarding the expression of the protein, https://doi.org/10.1038/srep37885. We included a sentence about the protein expression in the revision, citing our previous work.

  • The main mass-spectrometry experiment, possibly, made in one replicate. There is no information about replicates and the reproducibility of the analysis. However, the protein composition of matrix vesicles should depend on cell proliferation, their apoptosis status, and confluency and may vary between replicates. That's why it's crucial to repeat the experiment several times and analyze only reproducible proteins. Also, it's important to check the cell viability and proliferation upon TRIP1-overexpression.

We agree with the reviewer’s comments. The limitation of the number of MVs obtained from the cell culture restricted us from working with biological replicates. However, for our next work, we performed a TMT-based quantification study on the same sample and end up with similar results.

  • Another important issue regarding mass-spectrometry experiments is that there is no experiment on intracellular protein composition. The protein composition of matrix vesicles (MV) strongly depends on intracellular protein level and should be analyzed simultaneously with such data.

We agree with the reviewer that the intracellular protein has a great influence in the matrix vesicle composition. However, our attempts to identify intracellular proteins always ended up in mostly cataloging higher abundant cytoskeletal proteins such as actin, tubulin, myosin etc. We were not able to see any calcifying or mineralization proteins due to the hindrance by abundant proteins. We are still strategizing to minimize the effect of abundant proteins.  

The manuscript also contains a lot of minor issues that should be addressed:

  • No replicates and statistical analysis in Nanoparticle tracking analysis and TEM imaging.

Regarding TEM imaging and nanoparticle tracking systems, this is qualitative data and not meant for quantitative purposes. We clearly mentioned this in the revised version so that we don’t mislead the scientific community. We thank the reviewer for their suggestion.

  • In the qPCR experiment statistical analysis did not perform for GM vs GM-OE pairs, which are discussed in the text.

Is the author referring to growth media and mineralization media?

  • Figure 7. The presentation of the data is not very demonstrative, especially in the case of decreasing RNA amount. The y-axis should be logarithmic for clarity.

  • Figure 3 is in the wrong place in the manuscript.

We agree with the reviewer’s comments. The necessary corrections were carried out in the revised manuscript as suggested by the reviewer.

  • Figure 1. There isn't a sequence alignment that is discussed in the text and figure capture.

      The phylogenetic tree is constructed using the sequence alignment and since it is from 61 sequences, we didn’t show it in the manuscript. However, if the reviewer insists, we can include this in the supplementary information.

  • Phosphosite Plus prediction of PTMs. These resource does not contain predictions, it accumulates information about PTMs that were detected in high-throughput or low-throughput experiments.

We agree with the reviewers and modified it in the revised manuscript.

  • Figures 2, 4, 6. The figure text is so small. It's impossible to read it.

We agree with the reviewer’s comments. The necessary corrections were carried out in the revised manuscript as suggested by the reviewer.

Reviewer 2 Report

In this manuscript author(s) has been identified new role for the TRIP-1 protein in calcified tissues which exemplified using in-silico as well as proteomics studies. Manuscript has good content in terms of results however it lacks good data representation like figures, data interpretation and explanation. Author (s) consider improving manuscript with major revision, and following concerns need to be addressed.   

1.     Figures are not in order, like Figure 1 placed after Fig 3.

2.     In 2.1 Phylogenetic analysis, sentences in the manuscript lines from 85 to 88 are vague, like “very few gaps”, “exceptionally conserved except nematodes” and “among plants and animals”. Author (s) should consider deriving more information by focusing more on data interpretation of results. 

3.     In 2.1 phylogenetic analysis section there is no mention of Figure 1.

4.     Also Figure 1 typo(s) are there and written twice. And if there is one figure there should not be any need to specify in sub section like (a) 

5.     Figure 1 legends are not explained well, for example if phylogenetic tree constructed from sequence alignment, what does it represent or what is the interpretation of these results.

6.     “Each color in alignment represents an amino acid” there is only one color? Author(s) should explore and described more “time lengths” “nodes” “scales” as represented in Figure 1.

7.     In section 2.2, author (s) started sentence predictions of functional domain without explaining the length of protein, and also there phosphorylation or ubiquitination posttranslational modifications are not explained well, where they are happening like in WD40 domain or in the middle region like in loop. Author (s) should focus on the describing results what is length of protein, how these domains are arranged, where modifications occurred? Also figure 2 is not I order in the manuscript.

8.     Many unclear statements: rather they are only results findings in one sentence without proper descriptions or interpretations like “Putative phosphorylation sites were also identified” it suggest what, where are these identified in protein sequence or in domains and what would be the functional relevance, (if any)?  

9.     Figure 2 resolution is not good and hard to see the amino acids and lousy arranged and also sub sections (a) (b) and (c) are not mentioned in figure legends.

10.  In section 2.3, tertiary structure modeling using ITASSER, although C-score range signified higher confidence but it did not tell about model stereochemistry. Therefore, models must be validated using Ramachandran plot and QMEAN, to confirm the feasible stereochemistry. And also author mentioned that in SMART database 5 WD40 domains were found but in modeling how many? it’s not described.

11.  Proteins WDr5 and G-protein Beta subunit has “similar structure closest in terms of functions” need to be cited.

12.  Figure 3 labeled with (a) and (b) but hasn’t been described, also how much % structural similarity imparts WDr5 and G protein beta subunit with TRIP-1.

13.  In “Ingenuity pathway analysis” paragraph Figure 4(b) has not mentioned or described.

14.  Figure 4 (a) representation of legends in box are not clearly visible.

15.  Figure 4 (b) “shortest path with 2 intermediate nodes” are not clear in the figure since it’s too crowded with number of proteins. Author (s) consider increasing zoom in figure to clearly show these paths and also durapatite; basic calcium phosphate crystal and calcium phosphate are not explained in the paragraph.

16.  Figure 5 image showing the MC3T3 and MC3T3 TRIPOE, number should have written in normal digit forms rather than exponential like “e+7 “ which is hard to compare with standard error or deviation values.

17.  Also representative images of TEM should have some arrow for better visualization for MC3T3 & MC3T3 TRIPOE and scale of image is not visible and hard to understand at which resolution scale image has been captured.

18.  In proteomics analysis author (s) did not mention the False Discovery Rate percentage for for label free protein analysis.

19.  Figure 6 is not well represented (b) Reactome pathway with two colored bar graph plot is not clear and understandable. And also not explained in figure legends.

20.  In gene expression analysis, Figure 7, what is GM and DM (only abbreviation is there) presented as osteogenic & angiogenic marker. And BMP2, GM -4hr showing large error bar. All bars have two star and it’s not mentioned which statistical test applied for calculation of p- values and even compared with which.          

Author Response

We immensely thank the reviewers and the editor for reviewing our manuscript titled "Insights into the structure and function of TRIP-1, a newly identified member in calcified tissues" and for their valuable and constructive comments. The reviewer’s constructive criticisms have helped us in improving the overall quality of the manuscript. As per the editor’s and reviewers’ constructive comments, the corrections were made in the manuscript keeping the track mode on and submitted for your kind perusal. We hopefully believe that we have addressed all the comments mentioned by the reviewers carefully and precisely. We wish to thank the reviewer for his invaluable comment that has strengthened our manuscript.

In this manuscript author(s) has been identified new role for the TRIP-1 protein in calcified tissues which exemplified using in-silico as well as proteomics studies. Manuscript has good content in terms of results however it lacks good data representation like figures, data interpretation and explanation. Author (s) consider improving manuscript with major revision, and following concerns need to be addressed.   

  1. Figures are not in order, like Figure 1 placed after Fig 3.

We thank the reviewer for catching that. Regarding the order of the figure, we submitted figures and text separately to the journal, and they assembled figures and text. We believe it will get fixed in the final version of the manuscript.

  1. In 2.1 Phylogenetic analysis, sentences in the manuscript lines from 85 to 88 are vague, like “very few gaps”, “exceptionally conserved except nematodes” and “among plants and animals”. Author (s) should consider deriving more information by focusing more on data interpretation of results. 

We agree with the reviewer and removed the vague sentences.

  1. In 2.1 phylogenetic analysis section there is no mention of Figure 1.

We quoted fig 1 now in the results section

  1. Also Figure 1 typo(s) are there and written twice. And if there is one figure there should not be any need to specify in sub section like (a) 

Thanks to the reviewer, we corrected the typo and removed the section in figure 1

  1. Figure 1 legends are not explained well, for example if phylogenetic tree constructed from sequence alignment, what does it represent or what is the interpretation of these results.

We agree with the reviewer that the figure legend is missing those details. However, we included those details in the results and methods section and don’t want to repeat them in the figure legend.

  1. “Each color in alignment represents an amino acid” there is only one color? Author(s) should explore and described more “time lengths” “nodes” “scales” as represented in Figure 1.

Regarding colors, is the reviewer referring to figure 2? If yes, the different colors in the figure 2a correspond to PTMs

  1. In section 2.2, author (s) started sentence predictions of functional domain without explaining the length of protein, and also there phosphorylation or ubiquitination posttranslational modifications are not explained well, where they are happening like in WD40 domain or in the middle region like in loop. Author (s) should focus on the describing results what is length of protein, how these domains are arranged, where modifications occurred? Also figure 2 is not I order in the manuscript.
  2. Many unclear statements: rather they are only results findings in one sentence without proper descriptions or interpretations like “Putative phosphorylation sites were also identified” it suggest what, where are these identified in protein sequence or in domains and what would be the functional relevance, (if any)?  

Answer to questions 7 and 8: We included the length of the protein in the results section. We agree with the reviewer on the PTMs are not being described well. EIF3i is a well-known protein in transcriptional regulation, and there were several reports on their PTMs. However, we reported EIF3i for the first time in their role in mineralization, and hence we were very careful not to associate any PTMs with mineralization. We have plans to carry out our future studies on PTMs and mineralization.

  1. Figure 2 resolution is not good and hard to see the amino acids and lousy arranged and also sub sections (a) (b) and (c) are not mentioned in figure legends.

We agree with the reviewer regarding the quality of the figure. The quality was compromised when it was added to the .doc file. Hope it will get fixed in the final version of the manuscript. We now mentioned the sections in the figure.

  1. In section 2.3, tertiary structure modeling using ITASSER, although C-score range signified higher confidence but it did not tell about model stereochemistry. Therefore, models must be validated using Ramachandran plot and QMEAN, to confirm the feasible stereochemistry. And also author mentioned that in SMART database 5 WD40 domains were found but in modeling how many? it’s not described.

We understand the reviewer’s concern about the model stereochemistry. However, our group doesn’t have any expertise in Ramachandran plot analysis, and we are looking for groups who can work in this aspect. Hopefully, we can add this data in the next revision or in our next work. In our modeling, we were able to retrieve the secondary and tertiary structures of the protein, not the domains.

  1. Proteins WDr5 and G-protein Beta subunit has “similar structure closest in terms of functions” need to be cited.
  2. Figure 3 labeled with (a) and (b) but hasn’t been described, also how much % structural similarity imparts WDr5 and G protein beta subunit with TRIP-1.

Answer to questions 11 and 12: In the discussion section, line 266 to 278 describes the function of WDr5 and GPSß and their role with citation

  1. In “Ingenuity pathway analysis” paragraph Figure 4(b) has not mentioned or described.

We have discussed figure 4 (b) in lines 132 and 133, but failed to mention it. Now, we have mentioned figure 4 (b) in the text.

  1. Figure 4 (a) representation of legends in box are not clearly visible.

  1. Figure 4 (b) “shortest path with 2 intermediate nodes” are not clear in the figure since it’s too crowded with number of proteins. Author (s) consider increasing zoom in figure to clearly show these paths and also durapatite; basic calcium phosphate crystal and calcium phosphate are not explained in the paragraph.
  2. Figure 5 image showing the MC3T3 and MC3T3 TRIPOE, number should have written in normal digit forms rather than exponential like “e+7“ which is hard to compare with standard error or deviation values. 
  3. Also representative images of TEM should have some arrow for better visualization for MC3T3 & MC3T3 TRIPOE and scale of image is not visible and hard to understand at which resolution scale image has been captured.
  4. In proteomics analysis author (s) did not mention the False Discovery Rate percentage for label free protein analysis.
  5. Figure 6 is not well represented (b) Reactome pathway with two colored bar graph plot is not clear and understandable. And also not explained in figure legends.
  6. In gene expression analysis, Figure 7, what is GM and DM (only abbreviation is there) presented as osteogenic & angiogenic marker. And BMP2, GM -4hr showing large error bar. All bars have two star and it’s not mentioned which statistical test applied for calculation of p- values and even compared with which.

   Answers to questions 14 to 20: We thanks reviewers for the suggestions to improve the figures. We modified the figures now. 

Reviewer 3 Report

This study focusses on revealing the structure of TRIP1 by applying bioinformatic tools and further elucidate its biological function by employing proteomics. The manuscript tries to make an important contribution to the field of biological mineralization. The findings are interesting and worth reporting. However, the manuscript suffers from a very limited investigation of the function of TRIP1 in osteogenic differentiation which makes the conclusions drawn in the manuscript potentially overstated. Here are the points which need to be addressed -

1.  Fig.7 needs attention, in the text authors are talking about BGLAP whereas the figure nowhere has any graph for it. I assume OCN is mistaken for BGLAP. Also, it would be more appropriate if the authors arrange their graphs in the same order as described in the text. Overall, the RT-PCR results can be presented and explained in a clearer way.

2.  To add more functional clarity authors can also check the protein expression levels of osteogenic and angiogenic markers in MC-MVs and TRIP1OE-MVs.

3. Furthermore, authors can do a functional assay like alkaline phosphatase assay (ALP) to assess osteogenesis in the presence and absence of TRIP1 to validate its role in the osteogenic differentiation of stem cells.

4.    Discussion section in the manuscript is too long. Authors can make it concise and add more insights to the results section.

5.  The authors forgot to quote figure 1 in the figure description (Results 2.1). Also, the order of figures is not correct.

Author Response

We immensely thank the reviewers and the editor for reviewing our manuscript titled "Insights into the structure and function of TRIP-1, a newly identified member in calcified tissues" and for their valuable and constructive comments. The reviewer’s constructive criticisms have helped us in improving the overall quality of the manuscript. As per the editor’s and reviewers’ constructive comments, the corrections  were made in the manuscript keeping the track mode on and submitted for your kind perusal. We hopefully believe that we have addressed all the comments mentioned by the reviewers carefully and precisely. We wish to thank the reviewer for his invaluable comment that has strengthened our manuscript.

This study focusses on revealing the structure of TRIP1 by applying bioinformatic tools and further elucidate its biological function by employing proteomics. The manuscript tries to make an important contribution to the field of biological mineralization. The findings are interesting and worth reporting. However, the manuscript suffers from a very limited investigation of the function of TRIP1 in osteogenic differentiation which makes the conclusions drawn in the manuscript potentially overstated. Here are the points which need to be addressed -

  1. 7 needs attention, in the text authors are talking about BGLAP whereas the figure nowhere has any graph for it. I assume OCN is mistaken for BGLAP. Also, it would be more appropriate if the authors arrange their graphs in the same order as described in the text. Overall, the RT-PCR results can be presented and explained in a clearer way.

Following reviewers’ suggestion, we changed BGLAP to OCN and text was modified based on the graph order.

  1. To add more functional clarity authors can also check the protein expression levels of osteogenic and angiogenic markers in MC-MVs and TRIP1OE-MVs.

We agree with the reviewer on protein expression levels. Unfortunately, we were not able to detect many osteogenic and angiogenic candidates in our proteomics study. This points out the fact that higher abundant proteins hinder the signal of scarce proteins in proteomics study.

  1. Furthermore, authors can do a functional assay like alkaline phosphatase assay (ALP) to assess osteogenesis in the presence and absence of TRIP1 to validate its role in the osteogenic differentiation of stem cells.

We appreciate the reviewers comment on functional assay. Our group carried-out several functional studies using the TRIP-1 in our earlier works https://doi.org/10.1038/srep37885. We have cited the work in the introduction part. 

  1. The discussion section in the manuscript is too long. Authors can make it concise and add more insights to the results section.

We agree with the reviewer. However, there is very few to no works that cataloged proteins in matrix vesicles. So, we trust discussing these proteins might provide valuable insights into their function in mineralization.   

  1. The authors forgot to quote figure 1 in the figure description (Results 2.1). Also, the order of figures is not correct.

We thank reviewer for catching that. We quoted fig 1 now in the results section. Regarding the order of the figure, we submitted figures and text separately to the journal, and they assembled figures and text. We believe it will get fixed in the final version of the manuscript.

Round 2

Reviewer 2 Report

Queries from 15 to 20 not addressed well. 

Author Response

Dear Reviewer,

Thanks for taking time to do a second round review. We sincerely apologize for not addressing the questions 15-20 individually. In fact, we made all the corrections following your suggestions in the previous round. English language editing was done as per your suggestion. 

  1. Figure 4 (b) “shortest path with 2 intermediate nodes” are not clear in the figure since it’s too crowded with number of proteins. Author (s) consider increasing zoom in figure to clearly show these paths and also durapatite; basic calcium phosphate crystal and calcium phosphate are not explained in the paragraph.

We agree with the reviewer. We highlighted the intermediate nodes in the figure.

  1. Figure 5 image showing the MC3T3 and MC3T3 TRIPOE, number should have written in normal digit forms rather than exponential like “e+7“ which is hard to compare with standard error or deviation values. 

Following the reviewer’s suggestion, we changed the numbers from exponential to digit.

  1. Also representative images of TEM should have some arrow for better visualization for MC3T3 & MC3T3 TRIPOE and scale of image is not visible and hard to understand at which resolution scale image has been captured.

We included the bigger font in the scale for better visualization and arrows were inserted near few matrix vesicles.

  1. In proteomics analysis author (s) did not mention the False Discovery Rate percentage for label free protein analysis.

We have mentioned false discovery rate in the method section. However, we just mentioned in acronym. In the current version we modified it.

  1. Figure 6 is not well represented (b) Reactome pathway with two colored bar graph plot is not clear and understandable. And also not explained in figure legends.

We agree with the reviewer. Since it doesn’t give more meaningful insights, we removed the reactome pathway.

  1. In gene expression analysis, Figure 7, what is GM and DM (only abbreviation is there) presented as osteogenic & angiogenic marker. And BMP2, GM -4hr showing large error bar. All bars have two star and it’s not mentioned which statistical test applied for calculation of p- values and even compared with which.

Following the reviewer’s suggestion, we included a full description of the GM and DM in the figure. Regarding BMP2, we agree we have large error bars, and we think removing just this data will disrupt the uniformity throughout this figure. The statistics used were described in the methods sections. It is tested between groups, post-hos Turkey’s test were carried out.

Reviewer 3 Report

The authors have made the necessary corrections and addressed all the questions raised earlier. 

Author Response

We thank the reviewer for agreeing to do the second round of review.